# LGBTQIA+ Adolescents’ Perceptions of Gender Tailoring and Portrayal in a Virtual-Reality-Based Alcohol-Prevention Tool: A Qualitative Interview Study and Thematic Analysis

**DOI:** 10.3390/ijerph20042784

**Published:** 2023-02-04

**Authors:** Christina Prediger, Katherina Heinrichs, Hürrem Tezcan-Güntekin, Gertraud Stadler, Laura Pilz González, Patricia Lyk, Gunver Majgaard, Christiane Stock

**Affiliations:** 1Institute of Health and Nursing Science, Charité–Universitätsmedizin Berlin, Corporate Member of Freie Universität Berlin and Humboldt-Universität zu Berlin, Augustenburger Platz 1, 13353 Berlin, Germany; 2Department of Health and Education, Alice Salomon University of Applied Science, Alice-Salomon-Platz 5, 12627 Berlin, Germany; 3Institute of Gender in Medicine, Charité–Universitätsmedizin Berlin, Corporate Member of Freie Universität Berlin and Humboldt-Universität zu Berlin, Augustenburger Platz 1, 13353 Berlin, Germany; 4SDU Game Development and Learning Technology, Mærsk Mc-Kinney Møller Institute, University of Southern Denmark, Campusvej 55, 5230 Odense, Denmark; 5Unit for Health Promotion Research, University of Southern Denmark, Degnevej 14, 6705 Esbjerg, Denmark

**Keywords:** LGBTQIA+, sexual and gender minorities, adolescents, virtual reality, alcohol prevention, gender, gender-sensitive, thematic analysis

## Abstract

Gender-sensitive interventions in alcohol-prevention that target adolescents often lead to binary tailoring for girls and boys. However, increased societal and legal recognition of sexual and gender minorities as well as research with this age group demand a broader understanding of gender. Therefore, the present study addresses the question of how interventions should be further developed to include sexual and gender diversity by exploring LGBTQIA+ adolescents’ perceptions of gender portrayal and gender-tailoring using *Virtual LimitLab*—a virtual-reality simulation for training refusal skills under peer pressure to consume alcohol. Qualitative interviews with 16 LGBTQIA+ adolescents were conducted after individual simulation testing. Using a thematic analysis with reflexive orientation, four themes were identified: Statements on *relevance of gender*, opinions on *tailoring-* and *flirting options*, and opinions on *characters*. Participants called for greater diversity representation among the *characters*, regarding gender identity and sexual orientation, as well as for representing, e.g., racialised peers. Moreover, participants suggested expanding the simulation’s *flirting options* by adding bisexual and aromantic/asexual options. Divergent views on the *relevance of gender* and wishes for *tailoring options* reflected the participant group’s heterogeneity. Based on these findings, future gender-sensitive interventions should conceptualise gender in a complex and multidimensional manner that intersects with further diversity categories.

## 1. Introduction

People who belong to sexual and gender minorities—including but not limited to individuals who identify as lesbian, gay, bisexual, transgender, queer, intersex, and aromantic/asexual (LGBTQIA+)—often face stigma and discrimination and thus additional health risks and health disparities [1,2,3,4,5]. Indeed, a higher prevalence of substance use has been found among LGBTQIA+ subgroups than among their non-minority counterpart populations [1,2,3,4,6]. These findings should be interpreted in the light of multi-faceted causation and as a possible consequence of minority stress rather than as a direct consequence of sexual orientation or gender identity [1,4,6].

Compared with other substances, alcohol is the most frequently used and misused substance among LGBTQIA+ subgroups [4] as well as among all adolescents in Europe [7], which renders alcohol prevention in these target groups a matter of public health concern. Prevention among adolescents is crucial as attitudes, consumption patterns, and drinking motives are formed during this period of life [8].

One option for enhancing prevention that targets young people is to integrate digital interventions and new media [9]. As such, virtual reality (VR) applications, are digitally simulated 3D environments that users can experience and interact with via head-mounted devices [10,11]. Since VR is increasingly often used in some areas of education (e.g., [12,13]) and treatment (e.g., [10,14]), it seems to also be promising for use in alcohol prevention. VR can promote learning in safe environments [15] via immersion and explorative, practice-based first-hand experiences [12] with increased interactivity compared with traditional learning methods [16]. To date, a few interventions have been developed that use VR for alcohol prevention among adolescents [17], including the VR component of the Australian alcohol education program *Blurred Minds*, the Danish *VR FestLab* application, and *Virtual LimitLab*, which is the German version of the *VR FestLab*. All these interventions target the training of refusal skills and aim to increase users’ resistance to peer pressure when offered alcohol. Research on these VR tools for alcohol prevention has demonstrated promising results in terms of both user engagement during development [15,18,19] and user experience [20]. One cluster-randomised controlled trial involving the Danish *VR FestLab* indicated that although refusal self-efficacy did not significantly increase, preventive tendencies were observed for girls, younger adolescents, and adolescents with low/medium levels of family affluence [21]. However, more research on virtual-simulation-based prevention is necessary in order to further develop this approach [17,21].

VR additionally offers opportunities for tailoring, for example, by using gender-specific avatars that reflect the user’s identity in a virtual environment [22] in order both to increase the user’s ability to identify with the avatar and to create gender-specific scenarios. Gender-specific interventions are being called for in alcohol prevention in order to address differences in consumption patterns between girls and boys [23,24]. This call has led to the development of specific interventions that use a binary understanding of gender and, in turn, may reproduce and reinforce stereotypes [25]. In addition, it might lead to the inappropriate homogenisation of within-group differences and exclude adolescents of gender minorities. A previous study conducted by our research group with girls and boys on *Virtual LimitLab* [17] indicated diverging opinions among the participating adolescents on binary gender-specific tailoring options. Participants raised awareness and questions as to how to better integrate peers with diverse gender identities into the simulation and additionally agreed with conceptualising sexual orientation in a way that is more than hetero-oriented [17]. This finding indicates the need to include LGBTQIA+ adolescents in further gender-sensitive research that takes into account these adolescents’ specific needs [26,27].

In light of this need as well as the increasing societal and legal recognition of sexual and gender minorities, research on the inclusion of gender and sexual diversity in generic prevention approaches is urgently needed. In Germany, increased societal recognition of these minorities can be seen, for example, in the monitored decline of open homophobia [28], whereas increased legal recognition can be seen, for example, in the legal recognition of the third gender option of “diverse” for intersex people in 2018 and in the explicit inclusion of transgender, intersex, and non-binary adolescents in the Child and Youth Welfare Law in 2021. Moreover, in health research, sexual and gender diversity have been increasingly often acknowledged, as illustrated, for example, by the integration of sexual and gender diversity into routine population-based surveys [29]. However, in terms of prevention, sexual and gender diversity have been less well-integrated into interventions; rather, gender is conceptualised binarily (i.e., girls and boys) or separately (i.e., for LGBTQIA+ adolescents). Accordingly, current research also calls for gender-sensitive and diversity-oriented perspectives in digital health promotion [30]. Nevertheless, limited knowledge exists concerning how the inclusion (rather than the separate conceptualisation) of LGBTQIA+ populations in generic alcohol interventions could take place while acknowledging the needs of this community, especially since digital alcohol prevention that is targeted towards this high-risk group is widely lacking [27]. Therefore, the aim of the present study is to explore the perceptions of LGBTQIA+ adolescents on gender tailoring and portrayal in a VR-based alcohol-prevention tool. In turn, results can serve both as a basis for integrating sexual and gender diversity into generic gender-sensitive prevention and as a basis for further development of gender-sensitive interventions in the field that have a broader understanding of gender.

In order to clarify the understanding of some terms used in this paper, short working definitions are provided below. However, it is important to note that discussion on terms and understandings is evolving, and the definitions used here are not meant to be taken as authoritative or prescriptive; rather, these definitions should be viewed as explanations as to how the terms are used in the conducted study. Throughout the present text, the term “gender” is understood as a sociocultural variable [31] that comprises more than two possible expressions and that thereby includes diversity as well as within-group heterogeneity. Moreover, gender is understood to be multi-dimensional (e.g., gender identity, gender role, gender expression or institutionalised gender) [32] and to interact with other diversity categories across an individual’s lifespan, all of which have an impact on this individual’s health. The terms “female”/“male” and “girl”/“boy” (and their respective plurals) are used to refer to gender. In line with the World Health Organisation’s definition of adolescents, participants are described as adolescents up to and including the age of 19 years even though 18 is the age of legal adulthood in the present study context of Germany. The term “LGBTQIA+” (i.e., lesbian, gay, transgender, queer, intersex, aromantic/asexual, and a “plus” for further inclusions) is used as an open collective term while acknowledging that individuals sometimes reject labels outright [33] and that self-definitions among young sexual and gender minorities increasingly often differ from the abbreviated labels [34]. Therefore, the term is viewed as an umbrella term for people who identify with sexual orientations and gender identities that lie outside of the hetero- and cisnormative matrix. The plus sign includes further sexual orientations (e.g., pansexual and omnisexual) and gender identities (e.g., non-binary, genderqueer, agender, and genderfluid) that are not explicitly listed and covers people born with physical sex characteristics that do not fit typical binary notions of female or male bodies (who may or may not identify with any gender or any sexual orientation within the LGBTQIA+ umbrella). It should be noted that the term LGBTQIA+ is neither all-encompassing nor authoritative, which is why the abbreviation LGBTQIA+ is chosen to include the collective term queer. In orientation towards Ogette [35], the word “*white*“ is written lowercase and in italics in order to mark it as a political and social construction that describes an otherwise unnamed, privileged position in a societal power asymmetry rather than actual skin colour, whereas the term “Black” is written capitalised and not italicised in order to express a political self-designation of racialised people rather than the colour as an adjective.

## 2. Materials and Methods

### 2.1. Research Design

A qualitative approach was chosen in order to account for the explorative character of the study. Individual semi-structured interviews were favoured over group methods in order to facilitate the recruitment process in a difficult-to-reach target group with conceivably sensitive content. Transcripts were analysed using thematic analysis with reflexive and inductive orientation towards Braun and Clarke [36]. Quality criteria for conducting [37] and reporting [38] these designs were compiled (see Appendix A: Consolidated criteria for reporting qualitative studies (COREQ), 32-item checklist) [38].

### 2.2. The VR Simulation Virtual LimitLab

The scope of the examined VR simulation is to train adolescents in a virtual environment to recognise and resist peer pressure situations and to consider consuming alcohol moderately. The simulation is based on the behaviour change wheel created by Michie et al. [21,39] and was co-designed by students, teachers, and alcohol-prevention practitioners with an interdisciplinary research team in Denmark from 2018–2019 [19,40]. Later, the simulation was overdubbed in German. The simulation uses a 360° filmed video that can be classified as a training world [41] and that is shaped by the viewpoint of the camera and thus does not enable users to move freely. Instead, interactivity is possible with the plot, which changes according to the user’s decisions at the end of each scene, when different behaviour options must be selected (e.g., drinking alcohol, playing beer pong, dancing, flirting, approaching other peers, or supporting drunk peers). Depending on the user’s individual choices and calculated blood alcohol concentration (BAC), the plot of the simulation changes. Users can navigate via head motion and can select behavioural choices by focussing on option buttons. When starting the simulation, the user sees a screen with the following question: “Who do you want to attend the party as?” The user must then choose a female or male gender, as represented by bathroom figures. A few scenes differ according to the chosen gender option, though most of the simulation was designed to be gender-neutral and is thus the same for both avatars. However, the chosen gender influences the BAC calculation by using the average body mass index (BMI) of a 16-year-old girl or boy as well as the number of alcoholic drinks consumed and the time that has passed since the user’s last drink. The simulation consists of three settings with positively and negatively conceptualised social feedback and role models: A pre-party, a teenager’s house party, and the morning after, when the user wakes up in their bed and reads mobile text messages. Positive experiences consist, for instance, of flirting or receiving positive feedback in the text messages. Adverse experiences include, for example, vomiting, not being able to flirt, blacking out due to intoxication, and receiving negative feedback the morning after.

### 2.3. Inclusion Criteria and Recruitment

LGBTQIA+ adolescents aged 15–19 years were the population of interest. In order to ensure more successful participant recruitment, the World Health Organisation’s definition of adolescents—which ranges from up to and including 19 years—was applied, which enlarged the defined target group of the simulation (15–18 years) by 1 year. An additional inclusion criterion was users’ self-reported ability to participate in the interview in the German language. In addition to their own informed consent, minors below the age of 18 years had to obtain informed consent from their parents. Information sheets for parents did not name the target group of LGBTQIA+ in order to avoid outing adolescents to their parents. In the recruiting material, potential participants were addressed as “queer teens, LGBTQIA+ adolescents, teens completely without label and allies” (individuals who support a social group of which they themselves are not a part) since identity categories frequently emerge, are sensitive, and change, especially during adolescence. In order to assure closeness to the LGBTQIA+ community, recruitment was realised via convenience sampling in LGBTQIA+ youth clubs, centres, open social programmes, sports clubs, associations, self-organised groups, a queer nightclub, and community networks, which shared the study invitation via Instagram and/or offered the researcher C.P. to present the study to adolescents in one of their meetings. In total, 26 networks and organisations in Berlin, Germany, were contacted. C.P. personally presented the study invitation at seven locations. Study information sheets were provided, and contact details were distributed in order that interested individuals could contact the study team themselves. In addition to convenience sampling, snowball recruitment was also used by inviting participants to share the invitation with their queer peers. Due to the heterogeneity of the target group, the exploratory design of the pilot test within a specific population, and the limited time for data collection, data saturation was neither targeted nor possible.

### 2.4. Data Collection and Processing

Semi-structured individual interviews were conducted after allowing participants to test the *Virtual LimitLab* simulation for 15–30 min, which was sufficient for one or two playthroughs. Upon the wish of the target group during recruitment, the data-collection form was extended to pair interviews. Interviews took place at the Institute of Health and Nursing Science, Charité—Universitätsmedizin Berlin as well as—at the request of participants—in collaborating youth clubs. Data collection was realised between January and August 2022 (with a pause from February–May due to COVID-19 restrictions at the research institution). All interviews were conducted by one researcher alone (i.e., C.P.), who is a research associate with a master’s degree in educational science and public health and who is experienced in conducting qualitative interviews through a prior study [17]. The researcher’s positionality was a *white*, middle-class, able-bodied, cisgender woman who had contacted institutions and presented herself as an ally of the queer community, former social worker, who was interested in research in the further development of gender aspects in health promotion and prevention. None of the interviewees were known to the researcher before data collection. After an introductory phase with the possibility to ask questions, a head-mounted display (Destek V5, Thinkline Technology LTD, London, United Kingdom) with a smartphone (Samsung Galaxy A21 devices, Samsung, Suwon, South Korea) inside of it was handed out for individual testing. An interview guide with open-ended questions was used to ensure that all topics of interest were raised and handled flexibly and adaptably in each interview. The guide was created following Helfferich’s [42] four steps of collecting, checking, sorting, and subsuming possible questions of interest, which resulted in five sections to be covered during each interview:General perceptions of the simulation;The perception of gender aspects;The assessment of current tailoring and gender portrayal;Potentially important categories other than gender; andSuggestions for improvement and further development.

The interviews were digitally audio recorded and transcribed verbatim by an external service, with whom a data-protection contract had been signed. Additionally, field notes were written after each interview, including observations and impressions. After each interview, participants were asked to fill out a short questionnaire that comprised age, type of school, whether the participant had experience with alcohol and with VR, and in orientation on Baumann et al. [43] open-ended questions on gender identity and sexual orientation, ethnical/cultural self-description, questions with categorical answering options on assumed external cultural/ethnical attribution (using the question “Are you usually perceived as a *white* German?”), Religion, and a text box for further information (if desired). Participants received the manuscript prior to publication in order that they could provide feedback on the findings.

### 2.5. Data Analysis

Data were analysed in orientation towards reflexive thematic analysis, as described by Braun and Clarke [33]. While thematic analyses can generally be described as a family of approaches “that seek to develop ‘patterns’ (themes, categories) across cases” [44] (p. 37), the term “reflexive”—used by Braun and Clarke—emphasises the subjective, situated, active, and (self-)critical questioning character of the researchers. This approach accounts for decisions as to what, why, and how research and data analysis are applied and executed and calls for outlining decisions in order to provide clarity on the practice and process. The following decisions about data analysis were made: Themes were understood to represent meaningful patterns that are supported by an organising concept across the dataset [45] that consisted of codes that were strongly related to one another and that were defined by their importance rather than by the quantity of the quotes. The analysis sought to provide a detailed account of one aspect—namely perceptions of and statements about gender aspects in the context of the VR simulation rather than providing a detailed description of the entire dataset. Interview content on general VR experience, content evaluation, and technical issues was not further analysed. As the participants’ perspectives on gender aspects were the interest in the study, a more data-driven, inductive orientation to coding was chosen. Nevertheless, the content of the transcripts was shaped by both the interview guide and the research question. Themes from a previous study with girls and boys [17] were consciously reflected and retained for the first inductive coding in order to catch possibly new patterns. Semantic coding was chosen over latent coding in order to capture the explicitly expressed meanings of the participants since the research question involved perceptions rather than explanations beyond what participants had stated.

In detail, the process of data analysis consisted of the following six phases [36], which were recursively applied and thus included returning to previous steps during the process:Familiarisation with the dataset: Initially, interview transcripts were re-read, re-heard, and corrected where necessary, and first ideas were noted down. Field notes taken after each interview were also re-read. This step was performed by C.P.Generating initial codes: C.P. chose three of “the richest” interviews (in terms of the depth of the assessed data and the interview time). Transcripts from these three interviews were inductively coded in parallel by two researchers (i.e., C.P. and K.H.) by broadly marking text passages while saving context, contradictions, and double occupancy in the passages for later discussion. Subsequent discussion was used to seek collaboration when reflecting on the understanding of the transcripts and the initial codes rather than to seek consensus on a final codebook [45]. After discussion of each of these three interviews, contradictions and open questions remained. The remaining 13 interviews were coded by C.P. exclusively. The subsequent steps were performed by C.P. in a later discussion along with K.H. MAXQDA software (2022) (VERBI Software GmbH, Berlin, Germany) was used for computer-assisted organisation.Generating initial themes: After all data had been coded, theme candidates were aggregated by organising codes on a higher, more abstract level in terms of shared topics, strong relation, and content coherency. Initial, different thematic maps were developed and discussed in order to illustrate how the initial themes and code candidates were related to one another.Developing themes: The initial themes were reviewed by checking for internal homogeneity and external heterogeneity in relation to the coded extracts. Therefore, coded extracts were re-read, some passages were re-coded, and some codes were combined or split, when necessary. A theme that did not address the research question was removed (general VR experiences), and code candidates were revised and combined. This step was performed by C.P. and led to a revised thematic map.Defining and naming themes: Definitions of themes were generated, and relationships between preliminary themes as well as between individual codes were checked. The generated revised thematic map was used to analytically undergo the coded extracts, and the adaption led to a finalised thematic map. Until this phase, discussion among the researchers (i.e., C.P. and K.H.) and reflection with an external qualitative research group (i.e., Qualitative Forschungswerkstatt, Charité) had been used to reflect on and refine the analysis and interpretation in order to meet the quality criterion of inter-subject comprehensibility [37].Writing the report: In order to prepare the article, themes and codes were ordered in a list. This order did not represent a hierarchy; rather, it represented a logical order that enabled the researchers to comprehensibly lead through the results. This list was then filled in with all corresponding statements for each code and sub-code in order to serve as a basis for writing the present manuscript.

### 2.6. Ethics

The present study was approved by the ethics committee of the Charité—Universitätsmedizin Berlin (file number EA2/154/21). Information on the study was provided orally and in written study information sheets during recruitment. Adolescents were informed that their potential participation was entirely voluntary and that they could decline to answer any questions and could withdraw from the study at any time. Informed-consent forms were distributed before the interviews. For participants aged younger than 18 years, consent was also obtained by their parents or legal guardians. All participants provided informed consent for participation in the study and for digital audio recording. Personal data were processed in accordance with the European General Data Protection Regulation and the Declaration of Helsinki. As an expense allowance, participants received a gift voucher worth EUR 30.

## 3. Results

### 3.1. Participants’ Characteristics

Overall, sixteen LGBTQIA+ adolescents were interviewed, fourteen of whom were interviewed in individual interviews and two of whom were interviewed as a pair along with another participant upon their request. The interview duration (without the simulation try-out) ranged from 18–77 min, with an average of 46 min.

The participants’ ages ranged from 15–19 years, with a mean of 16 years. The sample’s characteristics and heterogeneity are described in Table 1, with each column in random order.

### 3.2. Overview of the Thematic Map

A thematic map was formed and refined throughout the analysis. The final version—as graphically presented in Figure 1—illustrates the relationships between themes, codes, and sub-codes. Four themes were identified:Relevance of gender;Tailoring options;Flirting options; andCharacters.

The order of the themes does not represent a hierarchical structure; rather, it represents a logical plot for the description of the results from more general to more specific aspects. In this sense, the first theme (i.e., *relevance of gender)* encompasses identity- and orientational elements. In contrast, the second theme (i.e., *tailoring options)* and the third theme (*flirting options)* focus only on identity or on orientational elements, respectively. The last theme (i.e., *characters)* differs from this focus on the “self” (i.e., “Who am I in the simulation?”/“Who am I interested in/can I flirt with within the simulation?”) by shifting to the “others” in the simulation. The first theme (i.e., *relevance of gender*) comprises two codes: The *unimportance* and the *importance* of gender both in the simulation and in real party contexts. The latter code is further divided into the sub-codes of the *queer-specific*, *female-specific*, and *male*-*specific* significance of gender. The second theme (i.e., *tailoring options*) encompasses codes of possible tailoring options (i.e., *two options*, *three or more options*, or *no selection*) and *other options* for tailoring aside from gender. The third theme (i.e., *flirting options*) consists of statements on orientation, while the fourth theme (*characters*) involves statements on the other characters within the simulation. Interconnections between codes and sub-codes are represented by lines since these lines link *tailoring options* to the stated relevance of particular gender groups.

A detailed description of the themes, codes, and sub-codes is explained and illustrated with direct and indirect participant quotations (cited by a random letter attributed to each participant) in the following section in order to meet the quality criterion of empirical foundation [37].

### 3.3. Description of Themes

#### 3.3.1. Theme 1: Relevance of Gender

The first theme comprises perceptions and arguments on stating or rejecting the *relevance of gender* in the simulation or in real-life teenage party contexts. These perceptions and arguments are summarised in the respective codes of *importance* and *unimportance*.

Initially during the interviews, many participants stated that gender played no role in their experience of the simulation and that gender was “*not important at all*” (K) or was “*completely irrelevant*” (G) to the scenes and interaction. Similarly, some participants said that in their real-life experiences, gender is not important in their actual party behaviour. In addition, even if an avatar of a different gender was chosen in a second playthrough of the simulation, no differences were perceived. Other participants forgot or did not remember which avatar they had chosen. Reasons mentioned for the unimportance of gender were primarily grounded in the gender-neutral or mostly gender-neutral perception of the simulation. One explanation for this gender-neutral perception was that the language used was perceived as neutral and that no pronouns were used. Another explanation was that flirting was possible independently of the chosen gender option both with same- and other-gender characters, which is why some participants felt that gender identity did not matter in the simulation. For example, this sentiment can be seen in the following quote: “*It [Flirting] went both ways, anyway, and [gender] was therefore irrelevant*” (H). In addition, another participant supported the unimportance of gender by explaining that in VR, users do not see themselves. An additional reason given for the unimportance of gender was perceived societal change: According to one participant, at least for the younger generation, in urban contexts, gender is less relevant. Furthermore, the unimportance of gender was described by many participants as a wish or desired ideal state: In an equal society, these participants felt that gender should not play a role and that no difference in the scenes should therefore be conceptualised, as reflected in the following demand: “*Gender is not so pronounced. That’s also what I actually want because I do not fit into that system, anyway*” (G). Moreover, participants frequently expressed that the personal context (e.g., the type of peer group or age)—rather than gender as a category—matters in real-life parties. According to participants, it is more important for the simulation that the individual user is able to decide on the plot development than that scenarios are presented which are tailored to gender. For example, as one participant described:

*“That’s why I think the avatar selection is so unimportant. You wouldn’t need to pre-select [an avatar gender] because it’s more important how I feel right now, how I’m doing at the party, and how I see things through my eyes rather than which category I belong to”*.(G)

Another participant concluded that the first-person perspective and the users’ decisions on the plot were decisive when it came to identifying with the simulation figure by explaining: “*If you just play the way you play, you feel like yourself, anyway*” (O).

On the other hand, some participants felt that gender was relevant both in the simulation and at real parties. In general, the importance of gender was mentioned as a possible tailoring category for better reaching certain groups in order that “*people would feel like they are being better addressed*” (I). According to some participants, gender does matter, especially at real parties, and these participants mentioned that different genders lead to different experiences. Therefore, as one participant explained, the simulation should reflect gender-specific experiences:

*“In real life, we all don’t experience the same night at the same party. And I think that when you simulate something like that, I think it’s good to pay attention to what different people experience”*.(B)

Moreover, before advocating for the unimportance of gender, one participant later stated that tailoring the simulation to gender-specific experience could “*reflect reality*… *unfortunately*” (G). Participants mainly voted for a predominantly neutral design but wished to include some gender-specific scenes. Then, the interviewer asked for gender-specific experiences and ideas as to what represented a gender-specific experience within the simulation. Next, the following statements were further divided into the sub-codes of specific experiences for *male*, *female*, and *queer* adolescents.

The typical male experience at parties was seen in verbal expressions, especially among boys. This experience was described as “*bro-like*” (A) for hetero peers and mainly involved “*how to hit on someone*” (E). Furthermore, boys were regarded by one participant as being more prone to excessive alcohol intake, which led another participant to the idea of depicting a male-specific scene with aggression and fights in the simulation.

Similarly, the participants saw typical female-specific experiences via a specific form of interaction among girls—namely giggling. Aside from the verbal expressions and specific interactions among these two gender groups, the most crucial differences were seen in the frequently mentioned gender-related topic of sexual harassment. Parties were reported to be generally “*more dangerous*” (L) for girls, and in combination with alcohol consumption, harassment was considered to be especially frequent and was deemed a relevant topic. Some participants declared that boys could also be affected by sexual harassment, albeit less frequently, but when harassment did occur, boys were reported to experience an even stronger taboo than girls. Some participants concluded that harassment should be presented in a female simulation scenario and that alcohol prevention should be combined with the prevention of sexual harassment, as illustrated, for example, in the following quote:

*“I think it’s a good idea to combine [alcohol prevention and the prevention of sexual harassment] because many things happen under the influence of alcohol and drugs that you don’t want to happen. Also, from my own experience, I can say a lot about what can happen, again and again, at mainstream parties and everywhere else. At my [queer] parties, unfortunately, that is also an issue, especially under the influence of alcohol”*.(G)

At the same time, participants were unsure as to whether and how to depict harassment in the simulation since the display might send the wrong message, for example, by teaching girls to have less fun and to be less outgoing, or it might trigger re-traumatisation. Therefore, some participants suggested including trigger warnings and the option to skip a scene. One participant brought up the idea of using VR for consent education and of including positive examples of asking for consent.

A typical queer-specific experience in real life reported in many sections of the interviews was hostility towards queers. Negative experiences ranged from getting views that were perceived as strange or confused to being stared at and being asked inappropriate personal questions. Additionally, participants reported having been insulted (e.g., by being called a “*faggot*” or a “*fucking lesbian*” (G)), excluded from athletic locker rooms, and physically attacked. These adverse experiences were reported to have taken place both in school and in public as well as in private contexts with peers and parents alike. The need to always adapt and explain oneself was described several times. According to some participants, including hostility in the simulation would make it more realistic, as indicated, for example, by the following quote:

*“Well, there [in general] is a lot of hostility. And at a typical house party, I would definitely expect some of that hostility. And if you want it [the simulation] to be close to reality, that [hostility] would have to be included”*.(I)

#### 3.3.2. Theme 2: Tailoring Options

The second identified theme was named *tailoring options* and comprised statements on customising the simulation with different avatar options, different depictions of these avatars, and different consequences for the following scenarios. Codes among this theme included perceptions regarding the current design of the simulation, in which a female or male avatar—represented by bathroom figures—had to be chosen before entering the simulation (*two options*) as well as ideas about and reflection on further options of tailoring (i.e., having *three or more options*, *no selection*, or *other options*).

The existing version of gender-tailoring by a female and male option was perceived differently by the participants. Some participants stated that they had chosen the avatar with which they identified and that they had perceived themselves as being this particular gender during the simulation. In contrast, predominantly negative perceptions were mentioned regarding the choice of a bathroom figure, which was described as “*difficult*” (N, H), “*not so great*” (P), “*stupid*” (I), “*restrictive*” (D), “*clich*é” (P), and “*violent*” (F). Participants additionally expressed feelings, such as “*I hate it [the binary choice] so much. It always scares me*” (O) and “*somehow, I felt tricked*” (A). For some participants, this led to the feeling of not being addressed. Moreover, the scene in which the choice had to be made was perceived as taking place too fast, which led some participants to not be able to recall which option they had chosen. Other participants reported having no particular feelings about this scene and described it as being “*no problem*” (A, K) and “*quite normal*” (M), whereas others viewed the scene with ambivalence and described it as “*sub-optimal, but not disturbing*” (J) and “*irritating, but [I’m] used to it*” (D). The choice was phrased as follows: “Who would you like to attend the party as?” This phrase was interpreted as a free choice, leaving room for trying out different genders and thus was rated positively by some participants as opposed to the question “What gender do you identify as?”.

Alternative ways of tailoring were brought up by participants or were asked for by the interviewer, including the idea of introducing more gender identities, which was summarised among the code of *three or more options*. Many participants mentioned the idea of introducing a third, fourth, or even further avatar gender options that could be selected prior to entering the simulation. Having more than a binary choice was considered to be more modern and to diminish the above-mentioned negative perception of the obligatory choice between girl or boy since the existence of more choices was viewed as “*being better for well-being*” (A). Another participant explained that adolescents who do not (exclusively) identify as a girl or a boy could be made to feel more included: “*That way, people would know, ’Yeah, there’s an option for me, too’*” (E). Participants were unsure regarding the accurate number, appropriate names, and representation (i.e., symbolic and/or textual) of additional gender-identity options. In fact, the implementation of three or more avatars was described as difficult, and respective terms were thought to possibly not be commonly known. Initially, some participants mentioned or favoured the idea of having a third avatar with the term “diverse”—which is the most well-known term in the German context—even if the participants personally disliked the term: “*I don’t like the word ’diverse’, but it’s what people would usually click on*” (F). At the same time, the term was discussed by participants as an often not-applicable collective term, as illustrated by one participant’s explanation:

*“Maybe ‘diverse’ [should be used] because it’s acknowledged by law as the third gender. But, of course, that’s not true because there are more than three genders”*.(J)

Many participants mentioned additional possible gender self-descriptions than these three. Other terms suggested were “*non-binary*” (B, D, L), “*genderqueer*” (A, F), and “*no gender*” (A). However, some participants stated that introducing this variety would take too much time since the “*the list would be too long*” (J). Furthermore, according to participants, the choice would be superfluous if it did not result in in-simulation consequences. The same applied to the idea that one participant had for designing the choice of gender as a spectrum bar between the two poles of female and male; however, after further reasoning, this idea was rejected since it was found to be sub-optimal, technically challenging, and not adequate if it was reduced in the end to three different scenarios. Some participants who advocated for more than three avatars thus finally concluded that it might be better to leave out the choice entirely, as summarised in the code of *no selection*.

Some participants brought up the idea of having *no selection* for avatars by leaving out the choice entirely. This was thought to lead to no tailoring during the simulation and was favoured by most participants when explicitly asked. A few counter-arguments that were mentioned included the idea that different experiences according to different genders should be addressed in the simulation and that trying out other gender avatars would no longer be possible without a selection option. Regarding the scenes that are presented later in the simulation, one participant doubted that the simulation could be appealing without a gender selection since the simulation might become too non-specific. In contrast, numerous positive descriptions of having *no selection* were mentioned, with some participants describing it as a better option compared with gender-tailoring and as being more direct, easy, and ideal. Some participants explained that the simulation was perceived mainly as being neutral and that gender was not at all or only marginally relevant to the simulation (cf. the code of *unimportant*). Furthermore, some participants wished for gender to be irrelevant when entering the simulation. Moreover, designing the scenarios in a neutral way was seen as a good option for not reproducing stereotypes, as illustrated by one participant:

*“Because if you differentiate between the scenes, it leads to stereotypes. And if, for example, you distinguish between… yeah, if you choose the male avatar, you get into a fight, and the girls end up on the toilet throwing up or something, then I think you’re really reinforcing stereotypes. So, I think it [the simulation] should be kept neutral”*.(L)

In addition, withholding possibly relevant scenes for certain genders was regarded as problematic. In summary, some participants stated that without a gender selection, they would have identified as themselves in the simulation, as clearly illustrated in the following participant statement: “*Because if there are no options at all, then I play as myself, anyway*” (O).

Furthermore, *other options* for tailoring that involved categories other than gender identity were mentioned or commented on, including:Building one’s own avatar with an individual look (which was considered exciting but overly complicated for a short simulation if this option did not have consequences within the scenes);Entering the user’s exact height and weight in order to personalise the BAC to a more precise body mass index (which was considered more realistic but a sensitive topic);Possibly including other relevant identity categories (which was considered irrelevant to the avatar choice in context of this simulation), and most prominently;Inclusion of a queer party scenario in the simulation.

Some participants felt that including a queer party scenario was a possible tailoring option. The current house party was described as being stereotypical, gender-conforming, and mainly *white*, as illustrated by one participant, who described the simulated house party as follows: “*[The simulation included] mainly white people with white names and very binary-looking peers (…) with straight couples*” (F). The simulation represented an event that some of the participants had never attended and would not even attend. One participant made the connection between the idea of having a queer scenario and the wish for a safer space:

*“But also, the same game doesn’t necessarily work for everyone. So, we cannot assume that a cis white male person would feel super safe at a BIPoC [i.e., Black, Indigenous, and People of Colour] queer party. I would also find it okay that it’s not like that. That’s also allowed to be, because we experience things much more often the other way around. Yeah, it would be interesting to have more options and to be able to choose a party”*.(F)

Therefore, participants were asked about what differences they saw between straight and queer parties as well as about how they would design a queer party scenario. The responses described queer parties as being basically the same as non-queer parties in terms of the types of interaction but as being more caring and socially inclusive and as having participants with greater social awareness and who give less importance to gender. However, the idea of a queer scenario was rejected by other participants since the simulation was intended to be the same for all simulation users. Moreover, upon reflection, some participants felt that having a queer scenario could be too different. Other participants suggested having a mix of a straight and a queer party by representing more gender diversity and possible orientations (which is represented further among the fourth theme of *characters*). A further possible version of tailoring that was mentioned involved changing the plot according to each individual’s actual behaviour by providing answer options from within the simulation that differed based on the level of alcohol intake. This option was described by many participants as being highly important and potentially effective for identifying with and becoming immersed in the simulation.

#### 3.3.3. Theme 3: Flirting Options

The third identified theme consisted of statements regarding the *flirting options* with other characters within the simulation. In the current simulation design, flirting is conceptualised as a positive alternative to drinking alcohol, and too much alcohol intake hinders flirting opportunities. Flirting options are mostly independent of gender appearance and the chosen avatar, and all characters are approachable or try flirting with the user, regardless of the chosen avatar gender. This openness is made explicit in a scene in which the user is asked by their peers whether and with whom they want to try flirting. The options are “girls”, “boys”, or “not currently interested in flirting”. This openness was mainly noticed and considered positive, with one participant describing it as “*surprisingly normal*” (G), especially as the party was perceived as a straight party. For example, another participant stated:

*“I just noticed that another male character asked me if I wanted to dance when I was playing as the male character. That’s really good because that’s so… I didn’t notice it at first because it was so normal. And that was really good”*.(O)

A point of critique of the current version of the simulation was that it retains the binary conceptualisation of either/or. Suggestions for improvements were made regarding the answering options in the above-described scene, in which the user is asked which gender is the desired one. Participants wished for more options, such as “*both*” (A, M) for bisexual and “*none*” (O, G, L) for aromantic/asexual users. Other users suggested adding the question “*What´s your type?*” (G), which could be designed to be less attached to gender. Another point of critique was related to the conceptualisation of flirting as being irritatingly fast and extreme, and more answering options between kissing and leaving were desired, such as continuing to talk with the person.

#### 3.3.4. Theme 4: Characters

Among the fourth identified theme, statements regarding the other characters and these characters’ appearance and behaviour as well as wishes and suggestions for interaction with these characters were gathered. Participants generally appreciated the fact that the actors were teenagers and that the simulation was filmed and not animated. Furthermore, they noticed that different typical characters were portrayed, such as “*the cool kids, the outsider, the funny guy playing beer pong, and so on*” (G), as one participant phrased it. On the one hand, this variety was described as being positive, and as being stereotypical and representative of clichés, on the other hand.

In general, many participants expressed the wish for more representation of diversity among the characters. Wishes included the representation of non-binary characters, characters with non-gender-conforming clothing, and couples that were gay, lesbian, or queer. Furthermore, the wish for more BIPoC peers as well as Black-, Arabic-, and Turkish-looking individuals and characters with headscarves was expressed. Some participants further explained that their wish was not to have “*a token black guy, like in advertisement, then I just don´t feel seen*” (D); rather, the wish was to normalise BIPoC main characters. Further suggestions included having physically disabled peers and characters with diverse economic family backgrounds. Additionally, home problems and psychological problems were mentioned as topics that could be represented in conversations since these issues were considered especially important in the context of alcohol prevention. Participants stated that by including these topics, they would feel personally addressed and more societally accepted. For example, one participant stated that having greater diversity representation would lead them to feel “*more comfortable and identified. That way, you could see that society is ready [for greater diversity]*” (H). All these suggestions were brought up in the interviews upon reflection. Some participants expressed the core wish of having greater representation in the simulation, whereas other participants expressed the desire for this representation, even if they felt that it was not a necessity, as illustrated in the following quote:

*“Me, as a non-German-looking person, you are somehow used to it [the lack of representation]. Basically, you already have it in your head as a standard that everyone is white. You don’t even pay attention to it anymore […]. So, the way [the simulation] was didn’t bother me. But it would also be a bonus if it [the simulation] were more diverse. And, of course, it’s always good to re-evaluate things. That is important. So that this standard way of thinking is not shaped. You can’t do it wrong, but better. And I would find it [the simulation] better with with more diversity. It would be a plus. Exactly. If there were more diversity, you might feel a little more seen. Not totally consciously, but perhaps subconsciously”*.(O)

The importance of representation can also be found in the positive perception of one character that had been designed to be an outsider at the party. This character was mainly viewed positively by several participants, who expressed that they could see themselves in the character, who was sitting alone.

One scene changed according to the chosen gender of the avatar, and the interviewer asked the participants explicitly about this scene as well as about one specific character in it. In the male-specific scenario, this character reacts to flirting by the user with the exclamation, “Hey, bro, I’m not gay. But we can have a beer.” This reaction was mostly perceived negatively by the participants. The character’s reaction was described as being “*offensive*” (N), “*disrespectful*” (C), “*homophobic*” (A), and “*anti-gay*” (F). However, the scene was considered good in terms of how realistic it was. According to the participants, hostility towards homosexuality is even harsher in reality, and in a straight party context, gay flirting would be even less well-accepted than lesbian flirting. One participant suggested removing this scene, whereas others suggested changing it to have a nicer tone. Some participants even developed the idea of offering counter-reactions, including providing positive social feedback to the user if the offensive character is verbally defied. Other participants developed the idea of designing a VR simulation to learn “*how to survive in an anti-homo world*” (O) and how to counter toxic masculinity for anti-discrimination purposes.

## 4. Discussion

Overall, the present study contributes to the research field of gender-sensitive alcohol prevention by including the feedback and evaluation of LGBTQIA+ adolescents. Four themes were identified by applying a reflexive-oriented thematic analysis. While the first theme (*relevance of gender*) comprised heterogeneous statements on the importance of gender, the other themes were characterised by two divisions; namely, the division between the self and others, on the one hand, and the division between identity and orientation, on the other hand. The latter division reflects the conflation of sexual orientation and gender identity under the umbrella term of LGBTQIA+, which encompasses both concepts. The second theme (*tailoring options*) comprised views on different options for customising the simulation with gender identities by including different avatars and scenarios, while the third theme (*flirting options*) encompassed perceptions on orientational aspects. In contrast, the fourth theme (*characters*) involved the views and wishes of the participants regarding the depiction of others within the simulation, with a clear desire for a greater representation of diversity having been found.

The participants were unified in their demand for characters that represent diversity in terms of sexual orientation and gender identity as well as in their desire for further diversity aspects, such as the inclusion of racialised peers, physically disabled peers, and characters with diverse economic and family backgrounds. Additionally, participants advocated for the representation of less visible issues—such as mental health—in the simulated conversations. These findings are in line with the literature, which has demonstrated the importance of representing role models in the media in order to facilitate LGBTQIA+ identity development [46]. In line with other studies, this finding points to the need for more complex, non-stereotypical depictions of gender that acknowledge the complexity and heterogeneity of different LGBTQIA+ subgroups [47,48]. Furthermore, this finding reflects the need for intersectional approaches to understanding and addressing diversity in health interventions [49].

In addition, the participants endorse that flirting options within the simulation should remain uninfluenced by the chosen avatar’s gender. Nevertheless, the simulation was found to utilise a binary depiction of gender, and participants made clear of their wish for optimisation by including bi- and aromantic/asexual orientations in the answering options. This finding is in line with Porta et al. [34], who demonstrated that sexual orientation labels among LGBTQIA+ adolescents exist on a continuum and are not only oriented towards “same” or “opposite” dimensions. Similarly, our previous study that examined *Virtual LimitLab* with non-LGBTQIA+ adolescents [17] indicated the desire to conceptualise sexual orientation in a way that goes beyond a heterosexual and binary understanding of gender.

In contrast, participants disagreed on the general *relevance of gender* both in the simulation and in real-life party settings. Similarly, in our previous study [17], no consensus was found on the *relevance of gender* or on *tailoring options* in the *Virtual LimitLab* simulation among the participants. While adolescents in both studies mentioned that gender was or should be irrelevant, gender was considered as a possible tailoring category that reflects different gender-specific experiences that exist in real life.

Therefore, the interviewer asked about gender-specific experiences in real life and about how the simulation could reflect these experiences. Participants of this study predominantly reported negative associations with queer-specific experiences, mainly of hostility that they had faced. Some participants proposed combining the topic of hostility with alcohol prevention in order to reflect reality and to address this issue in interventions. This proposal is consistent with our previous findings [17], which highlighted the wish to include sexual harassment as an important topic for girls. The present study adds support to the finding that gender-sensitive alcohol prevention could address the topic of hostility towards queer adolescents. Considering Guldager et al. [20], who found that peer pressure should be more overtly included in the Danish *VR FestLab* application, it would be important to consider intensifying negative gender-specific experiences in *Virtual LimitLab*. In this context, the suggestion of participants in the present study to include trigger warnings is highly relevant. Further research is needed to explore whether combining alcohol prevention with other interventional scopes is conducive to better addressing adolescents’ real-life gender-specific experiences.

The idea of using VR for anti-discrimination education—which was mentioned by our study participants—calls for future research. While several studies have suggested that perspective-taking in VR simulations can evoke and enhance empathy towards stigmatised groups (e.g., [50,51,52]), a recent article questioned VR as an “empathy machine” [53] (p. 10) when it comes to complex social issues and called for more research in this field. In addition, societal multi-level approaches are needed to address hostility towards LGBTQIA+ adolescents since digital approaches in public health have limited impact and do not solve complex social issues by seeking to support individuals in an effort to improve their skills in applications or simulations alone [54].

Furthermore, no consensus was observed among the study participants as to how the goal of including sexual and gender minorities could be realised by tailoring the simulation to particular avatars and scenarios. The current design of two options (i.e., female and male) was met with divergent reactions that ranged from approving to indifferent and mostly elicited negative emotions. Therefore, many participants mentioned or agreed with the idea of including further gender identities. However, the number and specific labelling of further gender identities was discordant. The introduction of a third option with the label of “diverse” was primarily mentioned by participants. This option reflects the introduction of the new administrative and statistical category of “diverse” as the third legally possible gender category in Germany in 2018. At the same time, participants considered this term to be problematic since as a collective term, it fails to reflect actual gender diversity. Therefore, the participants supported the idea of leaving out the selection entirely in order to not reinforce stereotypes or create an overly complex and time-consuming selection process for designers and users. Compared with Morgan et al. [55], who studied the role of avatars in gaming for trans- and gender-diverse youth, the present study found the same critique of binary avatars, but not the same importance of avatar customisation, which might be due to the short display time of *Virtual LimitLab*. As the results of Morgan et al. [55] indicate, avatar customisation might be of greater importance for longer serious games or for digital interventions with special benefits for gender-diverse adolescents.

Moreover, participants in the present study commented divergently on the idea of designing a queer scenario, which was described to have both advantages (e.g., by being more realistic and creating a safer space) and disadvantages (e.g., by being overly specific and therefore too exclusive). Instead, several participants stated that immersion was sufficiently achieved—as is already the case in the current version—by changing the plot according to the multiple-choice and behaviour options rather than by pre-choosing an avatar category. This finding regarding tailoring options is in line with the results of our previous focus group study on *Virtual LimitLab* [17], in which the adolescent users emphasised that the strength of VR lies in its individualised perspective and plot rather than in the avatars themselves. This does not indicate that gender and diversity should be neglected when tailoring the simulation, especially concerning gender-specific experiences, such as hostility towards queer adolescents. Instead, gender and diversity should be considered during technology development by applying tools from gender and diversity studies. For example the “Gender-Extended Research and Development” (GERD) model [56]—which poses reflective questions during each phase of digital application development—could be used.

Overall, most of the adolescents in the present study wished for the simulation to be designed in a gender-neutral way and appreciated the mostly neutral appearance of the current simulation. Considered together, the heterogeneity in terms of perceptions and suggestions among LGBTQIA+ adolescents is an important finding and indicates that this group cannot be conceptualised as homogeneous. This finding is in line with our previous study, which also found heterogeneity in the opinions presented by the participating girls and boys [17].

One of the main strengths of the present study is its inclusion of the difficult-to-reach group of LGBTQIA+ adolescents in the evaluation of a generic digital intervention. The diversity of sexual orientations and gender identities in the sample can be seen as a further strength of the study since this diversity enabled a broad variety of perspectives to be assessed. The interviews yielded rich insights into the research question of gender portrayal and tailoring and thus contributed to an emerging research field on the use of VR in alcohol prevention. At the same time, several limitations must be acknowledged.

First, due to the explorative qualitative design of the present study, its results cannot be generalised and should thus be restricted to the particular context and study sample. As no sample size was pre-defined and the time for data collection was limited, data saturation was neither targeted nor possible. Nevertheless, within the dataset of the sixteen conducted interviews, data saturation could be regarded as having been achieved in orientation towards Francis et al. [57] since no new codes could be identified in the final three coded interviews. However, it is not possible to rule out whether further recruiting would have added additional perspectives to the dataset.

Second, self-selection bias may have occurred due to the chosen convenience and snowball sampling in the study. These sampling strategies run the risk of resulting in homogeneous groups and thus in limited perspectives and results. However, the heterogeneous results contradict this risk of bias. While the broad and open use of the term LGBTQIA+ during recruitment can be regarded as acceptable, this use might have led to the limited participation of certain subgroups. None of the participants identified themselves with the label of gay or intersex. Although specialised organisations and youth groups for these two subgroups were contacted, none of these participants could be included in the sample. This led to an unbalanced representation of LGBTQIA+ subgroups, and the transferability of our findings to all LGBTQIA+ adolescents is thus limited. In addition, the majority of participants attended an upper-level secondary school or grammar school, assumed themselves to be usually perceived as *white* Germans, had no religion, and had experience with both alcohol and VR. Regarding further characteristics, other diversity categories, e.g., Muslim, Turkish, People of Colour, and adolescents with learning difficulties (as reported in the open-ended short questionnaire) were minimally represented. Another selection bias might have been the requirement of parental consent for participants aged below 18 years, which may have precluded the participation of some adolescents. Even though the potential of outing participants was avoided in parental information sheets, the ethical requirement of receiving consent by parents for minors may have impeded participation. Furthermore, recruitment was limited to Berlin, Germany, in order to guarantee in-person testing of the simulation as well as in-person interviews. Therefore, this study represents an urban, central European setting, and its results should be interpreted in light of this context.

Third, interviews may have been influenced by social desirability. Indeed, it is conceivable that participants were less critical or extra critical toward the virtual simulation. In order to counteract this issue, effort was made to emphasise that the development itself had been carried out by another research group (i.e., the Danish *VR FestLab* team) with the goal of ensuring greater openness among participants.

Finally, in light of the reflective and diversity-conscious nature of the present paper, it was written from the perspective of the authors, who are in a largely privileged position in terms of many characteristics, such as age, income, and societal position. However, reflecting on one’s own positionality is crucial in a qualitative research approach that does not deny subjectivity and rather uses this subjectivity as an inherently productive resource in research [44].

## 5. Conclusions

The present study highlighted the importance of moving beyond the binary conceptualisation of gender and of further considering other diversity characteristics in order to include adolescents in VR interventions.

Regarding gender portrayal, the central finding was the wish for a greater representation of diversity not only in regard to sexual orientation and gender identity, but also, for example, in terms of the inclusion of racialised peers. Regarding tailoring options, the study identified divergent positions among participants on a wide range of pre-selectable avatars and on the inclusion of a unique queer scenario. This finding highlights the importance of not conceptualising the LGBTQIA+ community as a distinct entity that has homogenous views and makes similar suggestions. Indeed, participants felt that “merely” adding a third avatar for sexual and gender minorities would not be sufficient. In fact, it might be best to tailor the studied simulation toward individual users via their actual behaviours rather than using a possibly stereotyped pre-selectable avatar gender. This idea would not represent a failure to acknowledge sexual and gender diversity in design and conceptualisation. However, based on our findings, for short prevention simulations, such as *Virtual LimitLab*, we suggest:Representing diversity (both in terms of gender and otherwise) among the characters;Inclusion of sexual diversity via open flirting options (i.e., not only homosexual options, but also bisexual and aromantic/asexual options); andConducting further research on whether and how to combine and address gender-specific experiences (e.g., hostility toward LGBTQIA+ adolescents) within alcohol-prevention interventions.

In this sense, acknowledging sexual and gender diversity indicates addressing the specific experiences of hostility toward LGBTQIA+ adolescents as part of a broader, multi-level approach since hostility toward these sexual and gender minorities is a far-reaching problem that is situated within societal power imbalances. In other words, more collective approaches are necessary. Inclusion of LGBTQIA+ adolescents in gender-sensitive prevention research is only a starting point in the research field.

Indeed, both LGBTQIA+ adolescents and the topic of gender need to be integrated into research, while maintaining a complex understanding of gender and sexuality that goes beyond binarity and reflects the intersectionality of diversity. Considering this, future digital strategies could thus result in interventions that are more inclusive of diversity-conscious approaches with and for all young people.

## Figures and Tables

**Figure 1 ijerph-20-02784-f001:**
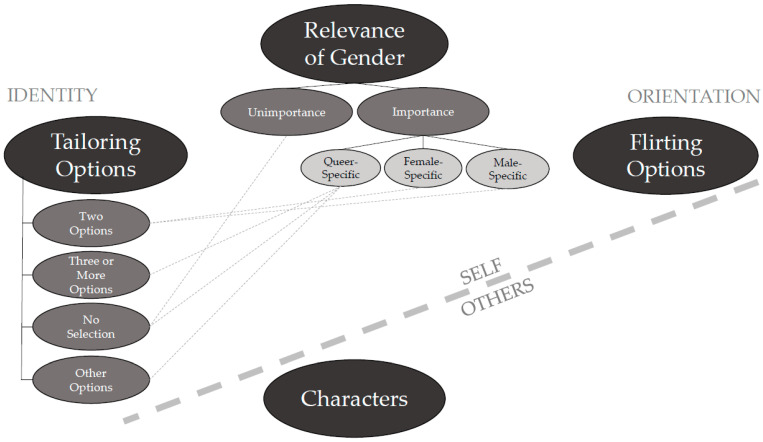
Final thematic map.

**Table 1 ijerph-20-02784-t001:** Participants’ characteristics (*n* = 16).

Age	Gender Identity	Sexual Orientation	Type of School	Ethnical /Cultural Self- Description	Are You Usually Perceived as a *white* German?	Religion	Experience with Virtual Reality	Experience with Drinking Alcohol
15–19 Years Mean: 16.7 years SD ^1^: 1.5	non-binary non-binary non-binary non-binary/genderqueer non-binary/diverse female female female cisgender women women/she women man male trans man trans man n. a. ^2^	queer/unlabelled queer aro/ace ^3^ bisexual/asexual bisexual bisexual bi pansexual pansexual lesbian no label no label in general I do not know trans n. a. n. a.	upper-level secondary school or grammar school (*n* = 12) integrated comprehensive secondary school (*n* = 2) vocational school (*n* = 1) school for children with learning difficulties (*n* = 1)	German (*n* = 8) *n*. a. (*n* = 4) *white* (*n* = 2) Turkish (*n* = 1) person of colour (*n* = 1)	yes (*n* = 11) no (*n* = 4) I do not know (*n* = 1)	none (*n* = 10) Muslim (*n* = 3) Catholic (*n* = 2) Protes-tant (*n* = 1)	yes (*n* = 12) no (*n* = 2) I do not know (*n* = 2)	yes (*n* = 14) no (*n* = 2)

^1^ SD: Standard deviation; ^2^ n. a.: No answer; ^3^ aro/ace: Aromantic/asexual.

## Data Availability

Data are not publicly available due to data protection.

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
