# Peer review of "LGBTQIA+ Adolescents’ Perceptions of Gender Tailoring and Portrayal in a Virtual-Reality-Based Alcohol-Prevention Tool: A Qualitative Interview Study and Thematic Analysis"

_ijerph, 2023, doi:10.3390/ijerph20042784_

Round 1
Reviewer 1 Report
The introduction, while generally very good and supportive of the need for the study, repeated some information. Please review to see if it could be more streamlined with shorter sentences (also true of the discussion).
The manuscript addresses the need for tailoring of alcohol prevention VR for gender nonconforming youth in Germany, a group that in some reports appear to have higher risk of drinking. the topic is original as it expands beyond the binary gender tailoring. I have not seen other research addressing this high risk group, so it is ground breaking.No comment or critique on the methods, the conclusions are consistent with findings presented and they address the research question. The references appear appropriate.
This latter comment extends to the rest of the manuscript. There were a few typos and misspellings (“whish”) but overall the science and contributions to the literature were strong.
Author Response
Dear Reviewer,
thank you very much for reviewing our article and for the helpful comments. Please find our responses interspaced in bold below and highlighted with track changes in the revised manuscript (one colour for content revision and another colour for language improvements by a native speaker).
Point 1/Repetitions in the introduction and discussion section
Comment: The introduction, while generally very good and supportive of the need for the study, repeated some information. Please review to see if it could be more streamlined with shorter sentences (also true of the discussion).
Response: Thank you for pointing this out. To be more precise, we rearranged the paragraphs in the introduction so that the research gap is described only once (lines 85-144). Also in the discussion we made rearrangements of the existing phrases within the limitations (lines 934-945. Further, to streamline the language, some sentences were rewritten or split after proofreading by a native English speaker.
Point 2/Corrections:
Comment: There were a few typos and misspellings (“whish”)
Response: Thank you, indeed there have been mistakes, which have been corrected after proofreading by a native speaker (applying for the whole manuscript).
Reviewer 2 Report
The paper is well written. I especially liked the section on the limitations of the research. The paper needs to be copy edited by a native/near-native speaker of English.
Author Response
Thank you very much for reviewing our article and for the helpful comment. Our manuscript is now revised and proofread by a native speaker represented by two different track-change colours for content and language improvements you can find in the manuscript.
Reviewer 3 Report
Thank you to the authors for their work on this paper. This is an interesting topic and the paper was engaging to read.
The comprehensive description of the methodological and ethical aspects of the study was terrific. I also appreciated the in-depth presentation and exploration of findings with quite extensive integration of participant voice.
Further line-specific feedback as follows:
** lines 54-55; an initial concise overview of existing prevention strategies would be helpful context here
** lines 94-121: thank you and well done to the authors for this comprehensive articulation of terminology/context/etc., this is very helpful and is the type of context that many articles ought to include.
** lines 671-675: this is unclear and needs re-writing/re-structuring for clarity
** lines 631-776 (discussion section): in depth and thoughtful with clear links to existing research and detailed acknowledgment of limitations, well done to the authors for their work here.
** lines 773-776: this is an important acknowledgment to include and I thank the authors for stating this; I wonder if it is worth fleshing out this section with further discussion around the approach you have taken (ethical, reflexive, intersectional) as I think this is worthy of consideration by other authors/researchers in this area (and in general, to be honest). Maybe worth commenting on best practice research ethics here? For example, mindfulness of one's own positioning, continued reflection, maintaining/sustaining a consciousness of diversity, privilege/marginalisation etc.
** lines 806-807: recommend rephrasing this slightly with a mind towards fully honouring the consciousness underpinning your project and also changing 'youths' to 'young people' which is more humanising, e.g. you could look at rewriting as: 'Considering this, future digital strategies could result in interventions which are more inclusive of and take diversity-conscious approaches with and for all young people'.
Thank you again to the authors for their work on this paper, I found it very interesting and insightful, and as a queer academic, I really appreciate the ethical consciousness that is apparent in their writing.
Author Response
Dear Reviewer,
thank you very much for the detailed review of our article and for the helpful comments. Please find our responses interspaced in bold below and highlighted with track changes in the revised manuscript (one colour for content revisions and another colour for language improvements by a native speaker).
Point 1/ Digital interventions and new media as a possibility to enhance prevention among young people
Comment: lines 54-55; an initial concise overview of existing prevention strategies would be helpful context here (“One option for enhancing prevention among young people is integrating digital interventions and new media.”)
Response: Thank you for the suggestion. To meet this point, we added a WHO document as a reference, supporting the statement that digital interventions can enhance and complement existing approaches. Further, this document provides a good overview of how to plan, develop and implement youth-centered digital health interventions (line 60).
Point 2/ Rephrasing for clarity
Comment: lines 671-675: this is unclear and needs re-writing/re-structuring for clarity.
(“When asked for gender-specific real-life experiences and how the simulation could reflect these participants of this study predominantly reported negative associations and experiences, especially hostility against queer adolescents, Some participants proposed to combine the topic of hostility with alcohol prevention to reflect reality and to address this issue in interventions”)
Response: We rephrased the sentences as follows (lines 833-838):
Therefore, the interviewer asked about gender-specific experiences in real life and about how the simulation could reflect these experiences. Participants of this study predominantly reported negative associations with queer-specific experiences, mainly of hostility that they had faced. Some participants proposed combining the topic of hostility with alcohol prevention in order to reflect reality and to address this issue in interventions.
Point 3/ Limitations: One’s own positioning
Comment: lines 773-776: this is an important acknowledgment to include and I thank the authors for stating this; I wonder if it is worth fleshing out this section with further discussion around the approach you have taken (ethical, reflexive, intersectional) as I think this is worthy of consideration by other authors/researchers in this area (and in general, to be honest). Maybe worth commenting on best practice research ethics here? For example, mindfulness of one's own positioning, continued reflection, maintaining/sustaining a consciousness of diversity, privilege/marginalisation etc.
(“Last but not least, in light of the reflective and diversity conscious nature of this paper, it was written from the authors’ perspective, a mainly privileged position regarding many characteristics, such as age, income, and societal position.”)
Response: Thank you. We have now added another argumentation and reference to this statement (line 965-968): However, reflecting one’s own positionality is crucial in a qualitative research approach that does not deny subjectivity and instead uses this subjectivity as an inherently productive resource in research [45].
Point 4/ Last sentence in the conclusion
Comment: lines 806-807: recommend rephrasing this slightly with a mind towards fully honouring the consciousness underpinning your project and also changing 'youths' to 'young people' which is more humanising, e.g. you could look at rewriting as: 'Considering this, future digital strategies could result in interventions which are more inclusive of and take diversity-conscious approaches with and for all young people'. (“Considering this, future digital strategies could result in interventions which are more inclusive for all youth.”)
Response: Thank you very much for supporting us with concrete phrasing. We changed the sentence according to your suggestion (lines 1006-1008): Considering this, future digital strategies could thus result in interventions that are more inclusive of diversity-conscious approaches with and for all young people.